# Case Series of *Listeria monocytogenes* in Pregnancy: Maternal–Foetal Complications and Clinical Management in Six Cases

**DOI:** 10.3390/microorganisms12112306

**Published:** 2024-11-13

**Authors:** Lucía Castaño Frías, Carmen Tudela-Littleton Peralta, Natalia Segura Oliva, María Suárez Arana, Celia Cuenca Marín, Jesús S. Jiménez López

**Affiliations:** 1Obstetrics and Gynecology Department, Hospital Materno-Infantil, Hospital Regional Universitario Málaga, Avenue Arroyo de los Ángeles S/N, 29011 Málaga, Spain; luciacastano@live.com (L.C.F.); carmenlittleton1997@gmail.com (C.T.-L.P.); nataliaseguraoliva@gmail.com (N.S.O.); cecuman2@yahoo.es (C.C.M.); jesuss.jimenez@uma.es (J.S.J.L.); 2Department of Surgical Specialties, University of Malaga, 29010 Málaga, Spain; 3Research Group in Maternal-Foetal Medicine Epigenetics Women’s Diseases and Reproductive Health, Biomedical Research Institute of Malaga (IBIMA), 29071 Málaga, Spain

**Keywords:** *Listeria monocytogenes*, pregnancy, listeriosis, chorioamnionitis, miscarriage, diagnosis of listeriosis, treatment of listeriosis

## Abstract

Background: *Listeria monocytogenes*, a Gram-positive intracellular bacillus, causes listeriosis, which is primarily transmitted through contaminated food and vertical transmission. The incidence of the disease is estimated to be between one and ten cases per million globally, with pregnant women being particularly vulnerable. Objective: The aim was to describe the clinical characteristics, management, and outcomes of cases of gestational listeriosis at Hospital Materno Infantil de Málaga in order to improve our understanding, diagnosis, and treatment of this disease. Methods: A retrospective analysis of six confirmed cases of listeriosis was performed in pregnant women or neonates. Diagnostic confirmation was achieved using analytic and microbiological methodologies, including blood cultures and the measurement of C-reactive protein (CRP). Ethical approval was obtained, and clinical data were reviewed for reasons related to consultation, treatment approaches, and complications. Results: The most frequently observed symptoms were fever and abdominal pain, with complications such as intrauterine death and suspected chorioamnionitis. Four patients were treated with antibiotics, leading to improved outcomes. No severe complications such as neurolisteriosis were observed. Conclusions: Pregnant women are more susceptible to *L. monocytogenes*, which can cause mild maternal symptoms but severe foetal outcomes, including prematurity, foetal death, or neonatal infections. Early diagnosis and prompt treatment are crucial for improving maternal–foetal outcomes.

## 1. Introduction

*Listeria monocytogenes* is a facultative intracellular Gram-positive bacillus responsible for listeriosis, an infection primarily transmitted through the consumption of contaminated food and vertically to the foetus [1,2].

This bacterium is found in ubiquitous environments. Its ability to survive in a wide range of pH, salinities, and temperatures and to form biofilms makes it resistant to a variety of adverse conditions. *L. monocytogenes* grows in the soil and is consumed by animals; it becomes a foodborne disease when humans eat contaminated food such as processed meat, unpasteurised dairy products, and uncooked fruits and vegetables. As it grows at refrigeration temperatures, it is capable of contaminating refrigerated food and raw meat. Therefore, proper food education should be provided to all pregnant women, advising them to avoid these high-risk foods. It is important to highlight that pasteurisation and most disinfectants eliminate the microorganism [1,3,4,5,6].

Following the ingestion of contaminated food, *L. monocytogenes* can be engulfed by gastrointestinal cells. This is achieved by the binding of a bacterial internalin to E-cadherin (CDH1) on the host cell [7], allowing the bacterium to enter the host without compromising the gastrointestinal tract’s integrity. LapB may also play a role in the process of adhesion and entry into the cell [7]. Once inside the cell, vacuolar escape is largely mediated by listeriolysin O, with the assistance of phospholipases PlcA and PlcB and a metalloproteinase called Mpl [8]. The bacterium undergoes rapid multiplication within the cytoplasm of the cell, and it is engulfed by neighbouring cells, spreading from cell to cell, without being exposed to antibodies, neutrophils, or antibiotics in the extracellular fluid. Bacterial motility within the cytoplasm of infected host cells and subsequent cell-to-cell spread is driven by actin polymerisation at the bacterial cell surface. A host actin monomer-binding protein called profilin and the bacterial surface protein are thought to be involved in this process [9]. The *L. monocytogenes* internalin surface protein and the human E-cadherin receptor also facilitate infection across the placental barrier [6].

As the bacteria exhibit an intracellular life cycle, host defences against *L. monocytogenes* rely on cell-mediated immunity. Therefore, any condition that weakens this response increases the risk of infection [10,11]. The role of immunomodulation during pregnancy is of paramount importance. Progesterone has been demonstrated to inhibit maternal immune responses at the utero-placental interface [6,12]. Its mechanisms include the suppression of pro-inflammatory responses; the inhibition of the activation of dendritic cells (DCs), macrophages, and natural killer (NK) cells; and a reduction in the production of pro-inflammatory cytokines such as TNF-α and IL-1β. In addition, IL-12, a cytokine that drives Th1 responses associated with inflammation and foetal rejection, is suppressed. Progesterone also inhibits the production of chemokines, including macrophage inflammatory protein-1α, macrophage inflammatory protein-1β, and RANTES, by CD8+ T lymphocytes. This reduces the recruitment of immune cells to the placenta and attenuates inflammatory responses. Therefore, while progesterone is essential for maintaining a tolerogenic environment necessary for foetal development, the same immunosuppressive effect may increase susceptibility to infections such as listeriosis [12].

The World Health Organization (WHO) reports that the global annual incidence of listeriosis ranges from one to ten cases per million people, with approximately 20% of cases resulting in neonatal infection. Listeriosis can manifest in the form of outbreaks or sporadic occurrences. It is estimated that the average person is exposed to this microorganism five to nine times a year. However, the implications and consequences of the infection depend on the patient’s immune status [4,10,13]. The incubation period is typically within the range of one to two weeks, although it can vary from a few days up to 90 days [13]. According to the WHO, there are two primary forms of listeriosis: the non-invasive and the invasive types. The non-invasive form affects immunocompetent individuals and presents as a relatively mild case of gastroenteritis. However, in susceptible populations, it can manifest as the invasive form, which can result in severe outcomes such as sepsis, meningitis, and a mortality rate of 20–30%. Groups at risk include the elderly, individuals with compromised immune systems, and pregnant women [2,10,13].

Listeriosis in pregnancy is generally defined as a clinical illness in the mother and/or child with isolation of *L. monocytogenes* from the mother, neonate, foetus, or placenta [5]. The incidence of listeriosis is 10 to 100 times higher than in the general population, with a reported incidence of 4–10/100,000 pregnant women per year in Europe and North America. This phenomenon accounts for approximately 20.7% of all cases worldwide [1,11]. The observed increase in incidence can be attributed to two main factors: firstly, the elevated detection of infection during pregnancy, and secondly, the heightened susceptibility to the disease due to a physiological suppression of cell-mediated immunity and a diminished gastrointestinal motility. It seems likely that these are a consequence of the increased progesterone levels that are observed during pregnancy [1,6,13,14].

*L. monocytogenes* has a strong affinity for placental tissue, enabling it to cross the placental barrier and cause infection in both the placenta and the foetus. It is estimated that over 80% of pregnant women infected with this bacterium experience significant complications for their foetuses or newborns [1,15]. Studies have reported a 65% risk of spontaneous abortion when infection occurs in the first trimester compared to a 26% risk in the second or third trimester [6].

The spectrum of clinical presentations of listeriosis ranges from asymptomatic patients to those presenting with flu-like symptoms such as fever, malaise, myalgias, and mild gastrointestinal symptoms. It may also manifest with non-specific obstetric clinical features, such as uterine contractions, abnormal foetal heart rate, labour, chorioamnionitis, and foetal loss. This nonspecific presentation, common to other conditions, presents a diagnostic challenge to obstetricians. Maternal symptoms may indicate a higher level of exposure to *L. monocytogenes* or increased susceptibility to the infection [1,2,5,6]. However, about 29% of maternal cases can be asymptomatic, which means that maternal symptoms alone cannot be considered a reliable predictor of adverse foetal effects [16].

While listeriosis in pregnant women can be mild, its neonatal consequences can be severe, including spontaneous abortion, preterm birth, and foetal death [1,5,14]. The incidence of neonatal listeriosis is approximately 8 per 100,000 live births. The increased severity in newborns is explained by their immune system deficiency [4,6,13,14]. In neonates, listeriosis can present in two distinct forms: early-onset and late-onset. Early-onset listeriosis manifests within the first week of life, typically as a consequence of transplacental infection, and may present as chorioamnionitis. Neonates often present with signs of sepsis, pneumonia, or meningitis, with a high prevalence of prematurity and mortality. It is estimated that over than half of pregnant women will develop an influenza-like illness. Furthermore, the pathogen can be isolated from maternal blood or the genital tract in 44–89% of cases [4,5]. On the other hand, late-onset listeriosis manifests after the first week of life and is typically associated with exposure to the pathogen at the birth canal [12]. These infants are usually born at term and initially appear healthy [4] but subsequently develop signs of meningitis. The mortality rate in late-onset infection is lower than that observed in early-onset cases [6]. In this form of listeriosis, the mother is often asymptomatic and it is often not possible to isolate the pathogen from maternal cultures [4]. The severity of early-onset listeriosis is influenced by the virulence factors of *L. monocytogenes*, including its ability to invade host cells, evade immune responses, and spread intracellularly [5,11], while in the case of the late-onset infection, its ability to persist in the environment, resist immune clearance, and adapt to the conditions of the neonatal host play major roles [5,11,17].

Diagnosis is challenging for obstetricians and is based on maternal or foetal clinical findings and bacterial detection within samples derived from maternal, foetal, or neonatal bodily fluids. According to several authors, culture is the most reliable approach during pregnancy. Consequently, a blood culture is recommended in the event of maternal fever above 38 °C and compatible clinical signs of listeriosis [2,5,6,13]. Once an infection has occurred, early detection is of the utmost importance, as timely antibiotic therapy may significantly reduce the risk of severe foetal complications associated with listeriosis [6].

With regard to treatment, there are currently no controlled trials that have established a definitive drug of choice or the optimal duration of treatment for listeriosis. However, the first-line antibiotics used are ampicillin and penicillin, both belonging to the beta-lactam family with proven safety during pregnancy and adequate transplacental passage [4]. These antibiotics are able to bind to PBPs and penetrate the cell, allowing them to reach adequate intracellular concentrations [11]. Ampicillin is regarded as the preferred option due to its superior pharmacokinetic profile and its ability to attain more effective concentrations in the amniotic fluid and in the foetal compartment, which is crucial for the treatment of maternal and foetal infections [4,11]. On the other hand, cephalosporins should not be used due to their lower effectiveness against this microorganism [6].

Although *L. monocytogenes* is susceptible to a wide range of antimicrobial agents and the incidence of antimicrobial resistance is low, some multidrug-resistant strains of *L. monocytogenes* have been isolated from various sources, including clinical samples, the food industry, and the environment. Since multidrug resistance was first documented in *L. monocytogenes* in France in 1988 [18], several mechanisms of antibiotic resistance have been described. Listeria has intrinsic microbial resistance to a number of compounds, such as broad-spectrum cephalosporins and monobactams, due to the low affinity of these drugs for PBP3, the enzyme that catalyses the final step of cell wall synthesis in *L. monocytogenes*. However, it can also acquire resistance through adaptive mechanisms. Acquired resistance mechanisms include target gene mutations, such as within genes encoding efflux pumps, or the acquisition of mobile genetic elements, including self-transferable plasmids and conjugative transposons [19].

The use of gentamicin along with ampicillin still remains controversial. Some in vitro studies suggest a synergistic effect when gentamicin is added to the treatment [20]; however, this effect is not seen in animal models [4,11]. Although there is little evidence supporting this treatment approach, some studies have shown a survival advantage with the combined therapy of gentamicin and a beta-lactam [15,21].

Antibiotic dosing is crucial to ensure adequate placental passage. In accordance with the recommendations set forth by the American College of Obstetricians and Gynecologists (ACOG), a typical regimen consists of administering 2 g of ampicillin every 6–8 h intravenously for at least 14 days. In cases where empirical treatment is initiated following known exposure in an asymptomatic pregnant woman, a dosage of 1 g of amoxicillin every 8 h for a period of 14 days is recommended [6].

For penicillin-allergic patients, the recommended treatment is trimethoprim-sulfamethoxazole (TMP-SMX), which is bactericidal against Listeria, achieves adequate levels in serum and CSF, and has been shown to be clinically effective in numerous studies [22]. However, in pregnant patients, this drug should be avoided during the first trimester, since it may interfere with folic acid metabolism, and during the last month of pregnancy, kernicterus should be avoided in the foetus [23]. In the event that neither a penicillin-based regimen nor TMP-SMX can be employed, the use of meropenem should be contemplated, with rigorous monitoring for efficacy, given that treatment failure has been demonstrated in previous cases [24].

Although the incidence of listeriosis is relatively low, the high morbidity and mortality rates in vulnerable populations make it a significant public health concern. However, early diagnosis is challenging due to the non-specific clinical presentation. To improve maternal and foetal outcomes, it is essential to raise awareness, implement prevention strategies, enhance diagnostics strategies, and develop new therapeutic approaches [10,14].

The objective of this retrospective study is to identify the most common reasons for consultation among pregnant women diagnosed with *L. monocytogenes* infection at our hospital, as well as to examine the resulting maternal–foetal complications. Furthermore, the aim is to develop a diagnostic algorithm to facilitate early detection of this infection and to make recommendations to ensure the most effective antibiotic treatment for pregnant women based on the recent literature.

## 2. Materials and Methods

A retrospective research study was conducted on a series of six clinical cases in pregnant women at the Maternal and Child Hospital of Málaga, part of the Regional University Hospital of Málaga, during the period between 2018 and 2023. This study was conducted in accordance with the Declaration of Helsinki and approved by the Ethics Committee of the CEI de Provincial Centre of Málaga, Spain (protocol code 1452-N-24 and date of approval date 14 September 2024).

A short series of six cases of confirmed maternal or neonatal infection by *L. monocytogenes* was selected, based on the detection of this pathogen in maternal, foetal, and/or neonatal biological samples. The inclusion criteria were pregnant women presenting clinical symptoms of infection by *L. monocytogenes* with microbiological confirmation. Cases of infections caused by other pathogens or those without diagnostic confirmation were excluded. A comprehensive analysis was conducted on these patients to ascertain the underlying reasons for their consultation, hospitalisation, clinical management, and potential maternal–foetal complications. This was performed with the objective of aligning our clinical experience with the available scientific evidence.

A literature review was performed using the UptoDate and PubMed databases, and a total of 24 relevant articles were selected. The reference management tool Zotero was used to organise and cite the literature included in the study.

## 3. Results

The following are the clinical cases of six Hispanic pregnant women with confirmed *L. monocytogenes* infection, as evidenced by the isolation of the bacterium from different samples cultured on blood agar.

Case 1: A 37-year-old woman, 22.2 weeks pregnant, presented to the emergency department with hypogastric pain and a 38.5 °C fever of two days. She was admitted with a diagnosis of fever of unknown origin. Further tests revealed no leukocytosis, a CRP level of 52 mg/dL, a negative urine culture, and a positive blood culture for *L. monocytogenes*. Antibiotic treatment was initiated (ampicillin 1 g/8 h for 10 days and gentamicin 240 mg every 24 h IV for 5 days, followed by oral amoxicillin for 3 weeks). Following clinical improvement, the patient was discharged. However, follow-up was lost due to transfer to another hospital.Case 2: A 35-year-old woman, 19 weeks pregnant, presented with hypogastric pain, vaginal bleeding, amniorrhea, and fever (38.5 °C). A foetal heartbeat was absent. She was admitted for the treatment of a late miscarriage and suspected chorioamnionitis. Antibiotic protocol (ampicillin 2 g/6 h + gentamicin 240 mg/24 h + clindamycin 900 mg/8 h IV) and vaginal misoprostol were administered. Additional diagnostic procedures revealed leukocytosis (25,610 cells/μL), elevated CRP levels (106 mg/L), and positive results for *L. monocytogenes* in both blood and placental cultures. A necropsy revealed histological findings compatible with a Listeria infection, namely bilateral acute bronchopneumonia with microabscesses.Case 3: An 18-year-old woman, 34.3 weeks pregnant, presented with uterine contractions, sensation of amniotic fluid loss, and light vaginal bleeding. The basic obstetric ultrasound examination yielded normal results. Upon digital examination, the patient was found to have a cervical dilation of 2–3 cm and cervical effacement. The patient was transferred to the delivery room for labour induction, with antibiotic coverage for an unknown beta-haemolytic streptococcus infection, treated with ampicillin. A healthy female infant weighing 2120 g was born with an Apgar score of 9/10. During hospitalisation, the newborn was diagnosed with *L. monocytogenes* infection. Subsequent tests revealed negative maternal blood cultures (performed after antibiotic initiation). However, the newborn’s throat, ear, and blood cultures were positive for *L. monocytogenes*. A lumbar puncture yielded negative results. The newborn was discharged after 13 days of antibiotic therapy (ampicillin 2 g/6 h and gentamicin 240 mg/24 h).Case 4: A 36-year-old woman, 38 weeks pregnant, was admitted from the Foetal Pathophysiology Clinic for foetal monitoring due to non-reassuring cardiotocography (foetal tachycardia and reduced variability) and maternal fever of 37.3 °C associated with general discomfort. She was transferred to the delivery room for induction due to suspected chorioamnionitis under antibiotic coverage (ampicillin 2 g/6 h and gentamicin 240 mg/24 h). The pregnancy ended in an urgent caesarean due to suspected foetal distress. A male infant was born with Apgar scores of 4/9/9, requiring respiratory support for distress. Subsequent diagnostic procedures revealed a positive maternal blood culture for *L. monocytogenes*, and placental histopathology indicated the presence of chorioamnionitis. Additionally, the newborn’s blood culture was positive for *L. monocytogenes*, although the lumbar puncture yielded negative results.Case 5: A 28-year-old woman, 22.5 weeks pregnant, presented with hypogastric pain and a temperature of 38.2 °C. A foetal heartbeat was absent on ultrasound. She was admitted for foetal demise and suspected chorioamnionitis. Antibiotic therapy (ampicillin 2 g/6 h + gentamicin 240 mg/24 h + clindamycin 900 mg/8 h IV) and vaginal misoprostol were initiated. Foetal expulsion and subsequent curettage proceeded without complications. Additional diagnostic tests revealed leukocytosis (20,520 cells/μL), CRP 99.5 mg/L, amniotic membrane culture positive for *L. monocytogenes*, and placental signs of chorioamnionitis consistent with listeria infection. The necropsy revealed histological findings compatible with a Listeria, including extensive visceral lysis precluding histopathological assessment, as well as acute focal pulmonary inflammatory infiltrate.Case 6: A 31-year-old woman, 29.2 weeks pregnant, was referred to the emergency department from an outpatient clinic for threatened preterm labour and suspected chorioamnionitis. Upon arrival, she presented with a fever of 38.1 °C, uterine contractions, and leukocytosis. Labour induction was initiated under antibiotic coverage (ampicillin 2 g/6 h + gentamicin 240 mg/24 h + clindamycin 900 mg/8 h IV), with foetal neuroprotection using magnesium sulphate and pulmonary maturation. The pregnancy was terminated by emergency caesarean section due to suspected foetal distress on non-reassuring cardiotocography. A female infant was born with Apgar scores of 5/7/10. Subsequent diagnostic procedures revealed leukocytosis (14,300 cells/μL), elevated CRP (52 mg/L), placental culture positive for *L. monocytogenes*, and placental signs of chorioamnionitis consistent with listeria infection. The newborn’s ear swab culture was positive for *L. monocytogenes*, while blood culture and lumbar puncture were negative.

### Interpretation of Results

Our incidence corresponds with the figures reported in the literature of 8 per 100,000 births. These clinical cases highlight the variability in the presentation of *L. monocytogenes* infection in pregnant women, ranging from fever with no clear focus to miscarriage and neonatal complications Table 1. Early recognition and appropriate antibiotic treatment are essential to improve maternal and foetal outcomes. Although this is a retrospective study with a limited number of cases, the findings highlight the importance of vigilant surveillance of at-risk populations and the need for care protocols that include diagnosis of listeriosis in pregnant women with compatible symptoms.

## 4. Discussion

This article presents a case series of gestational listeriosis. Infection with *L. monocytogenes* during this particular period is an important cause of obstetric and foetal complications. The incidence rate of listeriosis observed in our hospital, the Hospital Materno-Infantil of Málaga, between 2018 and 2023 is comparable to that documented in the scientific literature in Europe [1], with an incidence of 8 cases per 100,000 births.

As previously mentioned, the main mode of transmission is through contaminated food, which highlights the importance of education on proper food handling, especially in pregnant women [5,6]. However, the exact mode of transmission has not been identified in any of our patients. It is a common misconception that Listeria infections occur in specific outbreaks. In fact, sporadic cases are more common [6].

The majority of cases reported during gestation occur in the second and third gestational trimesters. This is supported by our case series, which shows that four women were infected during the second trimester, two in the third, and none in the first trimester. The most likely hypothesis for this is that *L. monocytogenes* infection in the first trimester is associated with early pregnancy losses that are not correctly diagnosed [1].

As previously stated, the range of maternal symptoms associated with listeriosis is extensive, encompassing those who are asymptomatic to those who present with influenza-like symptoms [5]. Our results demonstrated that the most prevalent symptoms among the mothers were fever and abdominal discomfort. Notably, there were no documented cases of neurolisteriosis or severe maternal sepsis. In view of the inherent complexity of diagnosing this infection, coupled with its non-specific clinical manifestations and low incidence, we have devised a potential diagnostic algorithm, which is presented in Figure 1. The objective of this algorithm is to facilitate an early diagnosis, primarily based on initial suspicion due to potential maternal food exposure, gastrointestinal symptoms, or the presence of fever. Such suspicion is corroborated by supplementary tests, including maternal blood cultures and determination of proinflammatory parameters at blood tests. The correct suspicion of Listeria chorioamnionitis allows for the early commencement of antibiotic therapy.

In the cases under consideration, maternal infection was identified on four occasions in placental culture, twice in maternal blood culture, and all three live-born infants exhibited positive biological samples for Listeria.

In regard to obstetric and neonatal complications among our patients, two cases of intrauterine death were diagnosed during the second trimester. In both cases, obstetric curettage was performed following foetal expulsion, and the placental culture yielded a positive result for *L. monocytogenes.* In both cases of stillbirths, necropsy revealed histological findings compatible with Listeria. In one case, there was severe bilateral acute bronchopneumonia with microabscesses. In the other, extensive visceral lysis prevented histopathological examination, with an acute focal pulmonary inflammatory infiltrate. Both placentas exhibited evidence of severe chorioamnionitis and acute funisitis.

Three pregnant women underwent labour induction due to suspected chorioamnionitis, of whom two resulted in preterm labour. In this group, there was one uncomplicated vaginal delivery and two caesarean sections due to cardiotocographic records indicating suspected loss of foetal well-being. Among these neonates, two had low Apgar scores. In both cases, *L. monocytogenes* was isolated from their blood, while in the other neonate, the otic exudate was positive.

The findings indicate that no malformations were reported in the live newborns. This finding is consistent with the existing literature that suggests early infection does not necessarily correlate with specific foetal anomalies [6]. However, it should be noted that none of the cases in our study had a first trimester infection.

Regarding treatment, the mothers in our series were treated empirically with ampicillin and gentamicin in accordance with the protocol for the management of chorioamnionitis at our hospital. The neonates were treated with ampicillin. No neonatal death was reported. This reinforces the notion that appropriate antibiotic therapy can significantly improve outcomes in cases of neonatal listeriosis [4,19]. The therapeutic management outlined in Figure 2 has been implemented with the objective of developing an effective strategy that mitigates the risk of complications for both mother and foetus while optimising therapeutic efficacy.

### 4.1. Study Limitations

We must acknowledge the limitations of our study. The lack of a control group and the small sample size may restrict the generalizability of the findings. Furthermore, the absence of follow-up in some cases prevents definitive conclusions regarding the evolution of complications.

### 4.2. Future Directions for Research

Future studies should focus on the efficacy of different treatments and the development of more effective prevention protocols, as well as explore listeriosis in other at-risk populations. The establishment of controlled, multicentre trials could enhance the understanding and management of this infection.

## 5. Conclusions

Pregnant women represent one of the highest-risk groups for developing listeriosis, accounting for approximately 20% of cases worldwide. Although the maternal prognosis is typically favourable, with symptoms that can be mild or even absent, the consequences for the foetus can be severe. This study demonstrates the association between listeriosis and adverse pregnancy outcomes, including stillbirth, preterm delivery, chorioamnionitis, an increased risk of caesarean section for suspected foetal compromise, and low Apgar scores. Listeriosis affects 8 in 100,000 newborns and, depending on the timing of infection, can present as meningitis, pneumonia, and neonatal sepsis, with a mortality rate of up to 20%.

Therefore, early diagnosis and treatment of this condition is crucial to improve pregnancy outcomes and reduce neonatal complications. A diagnosis of infection is based on isolation of the pathogen from normally sterile media, with a placental culture being the gold standard. The first-line treatment consists of high-dose ampicillin, given its ability to act at the intracellular level and its safety profile during pregnancy.

Thus, early detection and timely treatment are fundamental elements in improving maternal–foetal prognosis and reducing the morbidity and mortality associated with this infection. Ultimately, this will lead to improved health outcomes for both mothers and their infants.

## Figures and Tables

**Figure 1 microorganisms-12-02306-f001:**
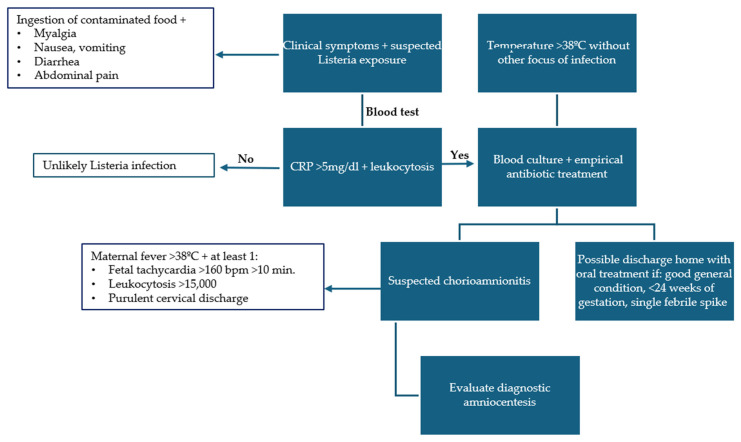
Diagnostic algorithm for *L. monocytogenes* infection (CRP: C-reactive protein).

**Figure 2 microorganisms-12-02306-f002:**
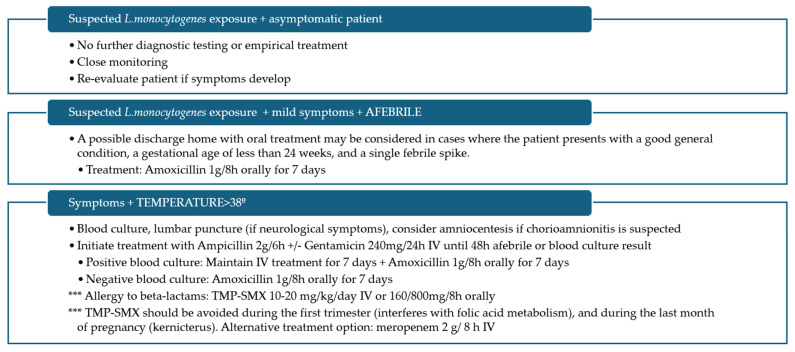
Therapeutic management of suspected listeriosis in pregnant women. (IV: intravenous, TMP-SMX: Trimethoprim-Sulfamethoxazole).

**Table 1 microorganisms-12-02306-t001:** Summary of six clinical cases of listeriosis in pregnant women (abbreviations: BC+: positive blood culture; D&C: dilation and curettage).

Case No.	Age (Years)	Gestational Week	Symptoms	Diagnosis	Treatment	BC+	Outcome
1	37	22.2	Hypogastric pain, fever 38.5 °C	Unknown origin fever	Ampicillin 1 g/8 h for 10 days; Gentamicin 240 mg IV for 5 days; Amoxicillin oral for 3 weeks	Yes	Discharged, lost to follow-up
2	35	19	Hypogastric pain, vaginal bleeding, loss of fluid, fever 38.5 °C	Late miscarriage, suspected chorioamnionitis	Ampicillin 2 g/6 h, Gentamicin 240 mg/24 h, and Clindamycin 900 mg/8 h IV; Misoprostol vaginal	Yes	D&C performed, positive necropsy for Listeria
3	18	34.3	Uterine contractions, amniorrhea, scant vaginal bleeding	Listeria infection in newborn	Ampicillin 2 g/6 h and Gentamicin 240 mg/24 h for 13 days	No	Normal delivery, infant infected
4	36	38	Maternal febrile illness, foetal distress	Chorioamnionitis	Ampicillin 2 g/6 h and Gentamicin 240 mg/24 h	Yes	Emergency C-section, neonate needed respiratory support
5	28	22.5	Hypogastric pain, fever 38.2 °C	Foetal demise, suspected chorioamnionitis	Ampicillin 2 g/6 h, Gentamicin 240 mg/24 h, and Clindamycin 900 mg/8 h IV; Misoprostol vaginal	Yes	Foetal expulsion, D&C performed
6	31	29.2	Fever 38.1 °C, uterine activity	Suspected chorioamnionitis	Ampicillin 2 g/6 h + Gentamicin 240 mg/24 h + Clindamycin 900 mg/8 h IV; Magnesium sulphate for neuroprotection	Yes	Emergency C-section, neonate infected

## Data Availability

The original contributions presented in the study are included in the article, further inquiries can be directed to the corresponding author.

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
