# Peer review of "Case Series of Listeria monocytogenes in Pregnancy: Maternal–Foetal Complications and Clinical Management in Six Cases"

_microorganisms, 2024, doi:10.3390/microorganisms12112306_

Round 1
Reviewer 1 Report
Comments and Suggestions for Authors
Listeria monoctogenes is a pathogen that poses a threat mainly among pregnant women, newborns and the elderly. Listeriosis mortality rates are high at around 20%. Undertaking research topics related to L. monocytogenes is important.
Comments to Authors:
- Listeria monocytogenes - please put in italics
- line 19 - change to listeriosis
- In the introduction, please complete the information about the occurrence of L. monocytogenes and the main source for humans. What is the antibiotic resistance of L. monocytogenes and epidemiological data on the infection rate in pregnant women and newborns
- line 64 - please use English spelling
- line 68 - please indicate the consent number of the relevant commission
- lines 69, 84, 146 - L. monocytogenes
- line 108 - please clarify, L. monocytogenes ?
- Table 1 - please remove the underline next to degrees C
- in the discussion, please standardize the spelling and position of the dots, in accordance with the introduction and the guidelines of the journal
- In the discussion section, too much information to be included in the introduction. I ask the authors to change the form of the discussion, i.e., juxtaposing their own results with other clinical cases
- line 244 - L. monocytogenes - written in italics
- Figure 1 - different font than main text - please correct
- Please explain under figure 1, any abbreviations that appear, including CRP
- Figure 2 - different font than main text - please correct
Author Response
Case Series on Listeria monocytogenes in Pregnancy: Maternal-Fetal Complications and Clinical Management in Six Cases.
Lucía Castaño Frías 1, Carmen Tudela-Littleton Peralta 1, Natalia Segura Oliva 1, María Suárez Arana 1,2,3, Celia Cuenca Marín 1,2,3 and Jesús S. Jiménez López 1,2,3
Response to Reviewer’s comments
|
Thank you very much for taking the time to review this manuscript. Please find the detailed responses below and the corresponding revisions and changes marked in red in the re-submitted files.
|
Point-by-point response to Comments and Suggestions for Authors |
Comment 1: Listeria monocytogenes - please put in italics |
Response 1: Corrected, thank you for pointing this out.
|
Comment 2: line 19 - change to listeriosis |
Response 2: Corrected, thank you. The change can be found in line 21 ´Methods: Six confirmed cases of listeriosis’
Comment 3: In the introduction, please complete the information about the occurrence of L. monocytogenes and the main source for humans. What is the antibiotic resistance of L. monocytogenes and epidemiological data on the infection rate in pregnant women and newborns Response: Thank you, we have included additional information in the introduction: - The main source for humans that can be found in line 43-51: `There are seven species of Listeria, and only four of the seven infect humans. Most infections are caused by serotypes 1A, 1B and 4B, and the last is responsible for the majority of the outbreaks of listeriosis (3).
This bacterium is found in ubiquitous environments. Its ability to survive in a wide range of pH, salinity and temperature and to form biofilms makes it resistant to a variety of adverse conditions. L. monocytogenes grows in the soil and is consumed by animals, it becomes a foodborne disease when humans eat contaminated food such as processed meat, unpasteurized dairy products, and uncooked fruits and vegetables. As it grows at refrigeration temperatures, it is capable of contaminating refrigerated food and raw meat. Proper food education should be provided to all pregnant women, advising them to avoid these high-risk foods. It is important to highlight that pasteurization and most disinfectants eliminate the microorganism (1,4–6).’
- Antibiotic resistant (lines 175-183): ‘Although L. monocytogenes is susceptible to a wide range of antimicrobial agents and the incidence of antimicrobial resistance is low, some multidrug-resistant strains of L. monocytogenes have been isolated from various sources, including clinical samples, the food industry and the environment. Since multidrug resistance was first documented in L. monocytogenes in France in 1988 (17), several mechanisms of antibiotic resistance have been described. Listeria has intrinsic microbial resistance to a number of compounds, such as broad-spectrum cephalosporins and monobactams, due to the low affinity of these drugs for PBP3, the enzyme that catalyzes the final step of cell wall synthesis in L. monocytogenes. However, it can also acquire resistance through adaptive mechanisms. Acquired resistance mechanisms include target gene mutations, such as within genes encoding efflux pumps, or the acquisition of mobile genetic elements, including self-transferable plasmids and conjugative transposons (18).’
- Epidemiological data on the infection rate in newborns in lines 132-135. ‘While listeriosis in pregnant women can be mild, its neonatal consequences can be severe, including spontaneous abortion, preterm birth, and fetal death (1,5,14). The incidence of neonatal listeriosis is approximately 8 per 100,000 live births. The increased severity in newborns is explained by their immune system deficiency (4,6,13,14). ‘
- Epidemiological data on the infection rate in pregnant women in line 102-108: `Listeriosis in pregnancy is generally defined as clinical illness in the mother and/or child with isolation of L. monocytogenes from the mother, neonate, foetus or placenta (5). The incidence of listeriosis is 10 to 100 times higher than in the general population, with a reported incidence of 4-10/100,000 pregnant women per year in Europe and North America. This phenomenon accounts for approximately 20.7% of all cases worldwide (1,11). ‘
Comment 4: line 64 - please use English spelling Response: Correction found in line 209.
Comment 5: line 68 - please indicate the consent number of the relevant commission Response: Added. The correction can be found in line 226-229 `The study was conducted in accordance with the Declaration of Helsinki, and approved by the Ethics Committee of the CEI de Provincial Centre of Málaga, Spain (protocol code 1452-N-24 and date of approval date 14 September 2024)’.
Comment 6: lines 69, 84, 146 - L. monocytogenes Response: Corrected, thank you. Lines 230-31; 247; 313-314 respectively.
Comment 7: line 108 - please clarify, L. monocytogenes ? Response: Clarification made, founded in line 270-271
Comment 8: Table 1 - please remove the underline next to degrees C Response: We apologise, but we are unsure how to interpret your statement. We have written the title of the table at the bottom in case you were referring to that.
Comment 9: In the discussion, please standardize the spelling and position of the dots, in accordance with the introduction and the guidelines of the journal Response: Thank you for your feedback. We have revised the discussion in an effort to provide a more structured overview of the symptoms, clinical course, diagnosis and treatment of our case series. We hope that these changes will enhance comprehension of the discussion. Given the extent of the revisions, we recommend a comprehensive review of the discussion.
Comment 10: In the discussion section, too much information to be included in the introduction. I ask the authors to change the form of the discussion, i.e., juxtaposing their own results with other clinical cases Response: Thank you for your feedback. We have removed the bibliographic information from the discussion and included it together with the new information suggested by the reviewers in the introduction. We have also expanded our commentary on the case series in the discussion. We hope you find the revised discussion satisfactory.
Comment 11: line 244 - L. monocytogenes - written in italics Response: Corrected, thank you for pointing this out. Line 389.
Comment 12: Figure 1 different font than main text - please correct Response: Corrected, thank you for pointing this out.
Comment 13: Please explain under figure 1, any abbreviations that appear, including CRP Response: Corrected, thank you for pointing this out.
Comment 14: Figure 2 - different font than main text - please correct Response: Corrected, thank you for pointing this out.
|
Additional clarifications |
We trust that you will find the revisions we have made appropriate. We have reorganised the structure of the information contained in the article, expanding the introduction with additional information we felt was relevant to the topic, and focusing the discussion on comparing our cases with the existing bibliography.
We have included the references below for your consideration:
- Charlier C, Disson O, Lecuit M. Maternal-neonatal listeriosis. Virulence. 31 de diciembre de 2020;11(1):391-7.
- Khsim IEF, Mohanaraj-Anton A, Horte IB, Lamont RF, Khan KS, Jørgensen JS, et al. Listeriosis in pregnancy: An umbrella review of maternal exposure, treatment and neonatal complications. BJOG Int J Obstet Gynaecol. 2022;129(9):1427-33.
- Posfay-Barbe KM, Wald ER. Listeriosis. Semin Fetal Neonatal Med. 1 de agosto de 2009;14(4):228-33.
- Lamont RF, Sobel J, Mazaki-Tovi S, Kusanovic JP, Vaisbuch E, Kim SK, et al. Listeriosis in Human Pregnancy: a systematic review. J Perinat Med. 25 de abril de 2011;39(3):227.
- Moran LJ, Verwiel Y, Bahri Khomami M, Roseboom TJ, Painter RC. Nutrition and listeriosis during pregnancy: a systematic review. J Nutr Sci. 2018;7:e25.
- Craig AM, Dotters-Katz S, Kuller JA, Thompson JL. Listeriosis in Pregnancy: A Review. Obstet Gynecol Surv. junio de 2019;74(6):362-8.
- Drevets DA, Sawyer RT, Potter TA, Campbell PA. Listeria monocytogenes infects human endothelial cells by two distinct mechanisms. Infect Immun. noviembre de 1995;63(11):4268.
- Faralla C, Rizzuto GA, Lowe DE, Kim B, Cooke C, Shiow LR, et al. InlP, a New Virulence Factor with Strong Placental Tropism. Infect Immun. diciembre de 2016;84(12):3584-96.
- Theriot JA, Rosenblatt J, Portnoy DA, Goldschmidt-Clermont PJ, Mitchison TJ. Involvement of profilin in the actin-based motility of L. monocytogenes in cells and in cell-free extracts. Cell. 11 de febrero de 1994;76(3):505-17.
- Schlech WF. Epidemiology and Clinical Manifestations of Listeria monocytogenes Infection. Microbiol Spectr. 17 de mayo de 2019;7(3):10.1128/microbiolspec.gpp3-0014-2018.
- Janakiraman V. Listeriosis in Pregnancy: Diagnosis, Treatment, and Prevention. Rev Obstet Gynecol. Fall de 2008;1(4):179.
- Ke Y, Ye L, Zhu P, Sun Y, Zhu Z. Listeriosis during pregnancy: a retrospective cohort study. BMC Pregnancy Childbirth. 28 de marzo de 2022;22(1):261.
- Raghupathy R, Szekeres-Bartho J. Progesterone: A Unique Hormone with Immunomodulatory Roles in Pregnancy. Int J Mol Sci. 25 de enero de 2022;23(3):1333.
- Listeriosis [Internet]. [citado 22 de octubre de 2024]. Disponible en: https://www.who.int/news-room/fact-sheets/detail/listeriosis
- Clinical features and prognostic factors of listeriosis: the MONALISA national prospective cohort study - The Lancet Infectious Diseases [Internet]. [citado 22 de octubre de 2024]. Disponible en: https://www.thelancet.com/journals/laninf/article/PIIS1473-3099(16)30521-7/abstract
- Allerberger F, Huhulescu S. Pregnancy related listeriosis: treatment and control. Expert Rev Anti Infect Ther. marzo de 2015;13(3):395-403.
- Frontiers | Different Shades of Listeria monocytogenes: Strain, Serotype, and Lineage-Based Variability in Virulence and Stress Tolerance Profiles [Internet]. [citado 22 de octubre de 2024]. Disponible en: https://www.frontiersin.org/journals/microbiology/articles/10.3389/fmicb.2021.792162/full
- Poyart-Salmeron C, Carlier C, Trieu-Cuot P, Courvalin P, Courtieu AL. Transferable plasmid-mediated antibiotic resistance in Listeria monocytogenes. The Lancet. 16 de junio de 1990;335(8703):1422-6.
- Antimicrobial Resistance in Listeria Species | Microbiology Spectrum [Internet]. [citado 22 de octubre de 2024]. Disponible en: https://journals.asm.org/doi/10.1128/microbiolspec.arba-0031-2017
- Azimi PH, Koranyi K, Lindsey KD. Listeria monocytogenes: Synergistic Effects of Ampicillin and Gentamicin. Am J Clin Pathol. 1 de diciembre de 1979;72(6):974-7.
- Sutter JP, Kocheise L, Kempski J, Christner M, Wichmann D, Pinnschmidt H, et al. Gentamicin combination treatment is associated with lower mortality in patients with invasive listeriosis: a retrospective analysis. Infection. 1 de agosto de 2024;52(4):1601-6.
- Winslow DL, Pankey GA. In vitro activities of trimethoprim and sulfamethoxazole against Listeria monocytogenes. Antimicrob Agents Chemother. julio de 1982;22(1):51.
- Treatment and prevention of Listeria monocytogenes infection - UpToDate [Internet]. [citado 28 de octubre de 2024]. Disponible en: https://www.uptodate.com/contents/treatment-and-prevention-of-listeria-monocytogenes-infection
- Stepanović S, Lazarević G, Jesić M, Kos R. Meropenem therapy failure in Listeria monocytogenes infection. Eur J Clin Microbiol Infect Dis Off Publ Eur Soc Clin Microbiol. junio de 2004;23(6):484-6.
Reviewer 2 Report
Comments and Suggestions for Authors
Obviously, the title is misleading, since this is not a comprehensive review of teh topic but a series of 6 cases wich are discussed in asimple way. This information can be obtaine din any textbook about the clinical problem of connatal listeriosis. A scientific, profound explanation about the virulence factors of L. monocytogenes, such as inlP (Faralla C et al, Infect Immun 2017) as well as an explanation about the pathophysiology (Vazquez-Boland et al., mBio 2017) are lacking.
line 44: tropism by experessing inlP.
line 46: severe maternal infection? in most cases the infected mother presents a short-lived, flu-like infection
line 51: food education: which to avoid? which to eat?
line 64: spanish
line 69: 6; only a short series
line 190: banal; by the way occasionally L. monocytogens can be transfered to the neweborn in the delivery room and not from the mother
line 200-202: repetition
line 203-212: what about the role of virulence factors of pathogenic bacteria
line 233: diagnostics? what about PCR?
line 233-238: the role of a combination of ampicillin with gentamicin in patients????? no profit! synergy is not seen in patients
line 233-238: therapy before diagnostics
line 238: no neonatal daeth"
line 240: "no neonatal death"
line 240: daignostics: the role of serology?. the role of multiplex PCR?
Fig. 1: Rather in practicable: Mother or child? Clinial symptoms and exposure to risk food should be separated. Fever alone is no good marker; in case of suspected chorioamionitis and amniocentesis: no bacteriologicic examination??
line 259: The penicillin binding proteins of Listeria do not bind cephalosporins!
Author Response
Case Series on Listeria monocytogenes in Pregnancy: Maternal-Fetal Complications and Clinical Management in Six Cases.
Lucía Castaño Frías 1, Carmen Tudela-Littleton Peralta 1, Natalia Segura Oliva 1, María Suárez Arana 1,2,3, Celia Cuenca Marín 1,2,3 and Jesús S. Jiménez López 1,2,3
Response to Reviewer’s comments
|
Thank you very much for taking the time to review this manuscript. Please find the detailed responses below and the corresponding revisions and changes marked in red in the re-submitted files.
|
Point-by-point response to Comments and Suggestions for Authors |
Comment 1: line 44: tropism by experessing inlP. |
Response 1: Thank you for your input. We included additional information which can be found in lines 54-68.
‘Following the ingestion of contaminated food, L. monocytogenes can be engulfed by gastrointestinal cells. This is achieved by the binding of a bacterial internalin to E-cadherin (CDH1) on the host cell (7), allowing the bacterium to enter the host without compromising the gastrointestinal tract's integrity. LapB may also play a role in the process of adhesion and entry into the cell (7). Once inside the cell, vacuolar escape is largely mediated by listeriolysin O, with the assistance of phospholipases PlcA and PlcB and a metalloproteinase called Mpl (8). The bacterium undergoes rapid multiplication within the cytoplasm of the cell, and it is engulfed by neighboring cells, spreading from cell to cell, without being exposed to antibodies, neutrophils, or antibiotics in the extracellular fluid. Bacterial motility within the cytoplasm of infected host cells and subsequent cell-to-cell spread is driven by actin polymerisation at the bacterial cell surface. A host actin monomer-binding protein called profilin and the bacterial surface protein are thought to be involved in this process (9). The L. monocytogenes internalin surface protein and the human E-cadherin receptor also facilitate infection across the placental barrier (6).’
|
Comment 2: line 46: severe maternal infection? in most cases the infected mother presents a short-lived, flu-like infection |
Response 2: Thank you for your comment. We have, accordingly changed this information, the symptomatic spectrum can be found between lines 121 and 134.
Comment 3: line 51: food education: which to avoid? which to eat? Response 2: Thank you. We added the following information about Listeriosis transmission and food education that can be found between lines 44 and 52
`This bacterium is found in ubiquitous environments. Its ability to survive in a wide range of pH, salinity and temperature and to form biofilms makes it resistant to a variety of adverse conditions. L. monocytogenes grows in the soil and is consumed by animals, it becomes a foodborne disease when humans eat contaminated food such as processed meat, unpasteurized dairy products, and uncooked fruits and vegetables. As it grows at refrigeration temperatures, it is capable of contaminating refrigerated food and raw meat. Proper food education should be provided to all pregnant women, advising them to avoid these high-risk foods. It is important to highlight that pasteurization and most disinfectants eliminate the microorganism (1,4–6).’
Comment 4: line 64 - Spanish Response 4: Corrected, thank you. Changed to “Materials and Methods” in line 223.
Comment 5: line 69: 6; only a short series Response 5: Thank you. Changed to “A short series of six cases” (line 231).
Comment 6: line 190: banal; by the way occasionally L. monocytogens can be transferred to the newborn in the delivery room and not from the mother Response 6: Agreed, we have omitted that sentence
Comment 7: line 200-202: repetition Response 7: Corrected, thank you.
Comment 8: line 203-212: what about the role of virulence factors of pathogenic bacteria Response 8: Thank you for your input. We have included additional information regarding this topic which can be found in lines 54-68.
Following the ingestion of contaminated food, L. monocytogenes can be engulfed by gastrointestinal cells. This is achieved by the binding of a bacterial internalin to E-cadherin (CDH1) on the host cell (7), allowing the bacterium to enter the host without compromising the gastrointestinal tract's integrity. LapB may also play a role in the process of adhesion and entry into the cell (7). Once inside the cell, vacuolar escape is largely mediated by listeriolysin O, with the assistance of phospholipases PlcA and PlcB and a metalloproteinase called Mpl (8). The bacterium undergoes rapid multiplication within the cytoplasm of the cell, and it is engulfed by neighboring cells, spreading from cell to cell, without being exposed to antibodies, neutrophils, or antibiotics in the extracellular fluid. Bacterial motility within the cytoplasm of infected host cells and subsequent cell-to-cell spread is driven by actin polymerisation at the bacterial cell surface. A host actin monomer-binding protein called profilin and the bacterial surface protein are thought to be involved in this process (9). The L. monocytogenes internalin surface protein and the human E-cadherin receptor also facilitate infection across the placental barrier (6).
Comment 9: line 233-238: the role of a combination of ampicillin with gentamicin in patients????? no profit! synergy is not seen in patients Response 9: Thank you, we agree that is a controversial statement. The correction can be found between lines 188-193.
The use of gentamicin along with ampicillin still remains controversial. Some in vitro studies suggest a synergistic effect when gentamicin is added to treatment regimens based on the fact that ampicillin weakens the bacterial cell wall and facilitates the entry of gentamicin, which inhibits protein synthesis inside bacterial cells; however, this effect is not seen in animal models (11). Although there is little evidence supporting this treatment approach, some studies have shown a survival advantage with the combined therapy of gentamicin and a beta-lactam (20,21)
Comment 10: In the discussion section, too much information to be included in the introduction. I ask the authors to change the form of the discussion, i.e., juxtaposing their own results with other clinical cases Response 10: Thank you for your comment. We have removed bibliographic information from the discussion and included it together with new information suggested by the reviewers in the introduction. We have attempted to comment further on the case series in the discussion. We hope you find the revised discussion satisfactory. As so many paragraphs have been changed, the corrections can be found in the introduction and discussion.
Comment 11: line 233-238: therapy before diagnostics Response 11: Corrected, thank you. Information regarding the diagnostic approach can be found between lines 152-158 and that regarding treatment in lines 160-206.
Comment 12: line 238 "no neonatal death" Response 12: Correction of the spelling, line 382 thank you.
Comment 13: line 240 "no neonatal death" Response 13: Thank you, it has been deleted.
Comment 14: line 240: diagnostics: the role of serology?. the role of multiplex PCR? Response: Thank you for your input, we have included additional information about diagnostic, and now it's located in the introduction. The information can be found between lines 154-161.
Diagnosis poses a challenge for obstetricians and is based on maternal or fetal clinical findings along with bacterial detection, which can be performed using PCR or cultures from maternal, fetal, or neonatal fluids. Culturing is the most reliable method during gestation according to various authors; thus, obtaining blood cultures in the presence of maternal fever exceeding 38ºC with compatible clinical signs of listeriosis is recommended (2,5,6,12)
Comment 15: Fig. 1: Rather in practicable: Mother or child? Clinical symptoms and exposure to risk food should be separated. Fever alone is no good marker; in case of suspected chorioamionitis and amniocentesis: no bacteriologicic examination?? Response: Thank you for your valuable feedback to this discussion. Each of your observartions will be addressed individually and we hope you find out comments of value.
Firstly, the algorithm refers to the maternal perspective, given that the review is centred on maternal infections and that our professional activities are primarily focused on pregnant women.
Furthermore, we have included cases of women with suspected exposure to contaminated food along with the corresponding clinical symptoms, in accordance with the recommendations set forth in the recent literature (e.g., as outlined by Uptodate). These recommendations outline the lack of need of further diagnostic testing in cases of exposure to potentially contaminated food in asymptomatic patients. Consequently, although these patients have not been incorporated into the diagnostic algorithm depicted in Fig. 1, we have included the approach in this cases in Fig. 2.
With regard to fever as an indicator of infection, we have employed it on the basis that the mother may present with a wide range of symptoms, none of which are specific for the diagnosis of listeriosis and which may vary considerably. In the MONALISA study, fever was reported in up to 65% of mothers, and thus we selected it as a potential screening marker. However, since it is rather unspecific, we include the use of analytical parameters that suggest a bacterial origin of the infection, such as leukocytosis with neutrophilia and elevated C-reactive protein. An elevation in CRP in conjunction with leukocytosis in the absence of other focality may support the necessity for a blood culture.
In cases where there is clinical suspicion of chorioamnionitis, amniocentesis should be performed for microbiological (Gram and culture) and biochemical studies. However, additional information regarding the diagnostic approach in case of suspected chorioamnionitis has not been included in the article, as it was considered to be beyond the scope of this investigation.
Comment 16: line 259: The penicillin binding proteins of Listeria do not bind cephalosporins! Response: Corrected, thank you. Lines 180-186 Listeria has intrinsic microbial resistance to a number of compounds, such as broad-spectrum cephalosporins and monobactams, due to the low affinity of these drugs for PBP3, the enzyme that catalyzes the final step of cell wall synthesis in L. monocytogenes. However, it can also acquire resistance through adaptive mechanisms. Acquired resistance mechanisms include target gene mutations, such as within genes encoding efflux pumps, or the acquisition of mobile genetic elements, including self-transferable plasmids and conjugative transposons (18).
|
Additional clarifications |
It is our hope that the revisions we have made will be deemed appropriate. The structure of the information contained in the article has been reorganized, with the introduction expanded to include additional information that we felt was relevant to the topic. This includes information regarding the virulence factors of L. monocytogenes and the pathophysiology of the infection. The discussion has been focused on comparing our cases with the existing bibliography.
We have included the references below for your consideration:
- Charlier C, Disson O, Lecuit M. Maternal-neonatal listeriosis. Virulence. 31 de diciembre de 2020;11(1):391-7.
- Khsim IEF, Mohanaraj-Anton A, Horte IB, Lamont RF, Khan KS, Jørgensen JS, et al. Listeriosis in pregnancy: An umbrella review of maternal exposure, treatment and neonatal complications. BJOG Int J Obstet Gynaecol. 2022;129(9):1427-33.
- Posfay-Barbe KM, Wald ER. Listeriosis. Semin Fetal Neonatal Med. 1 de agosto de 2009;14(4):228-33.
- Lamont RF, Sobel J, Mazaki-Tovi S, Kusanovic JP, Vaisbuch E, Kim SK, et al. Listeriosis in Human Pregnancy: a systematic review. J Perinat Med. 25 de abril de 2011;39(3):227.
- Moran LJ, Verwiel Y, Bahri Khomami M, Roseboom TJ, Painter RC. Nutrition and listeriosis during pregnancy: a systematic review. J Nutr Sci. 2018;7:e25.
- Craig AM, Dotters-Katz S, Kuller JA, Thompson JL. Listeriosis in Pregnancy: A Review. Obstet Gynecol Surv. junio de 2019;74(6):362-8.
- Drevets DA, Sawyer RT, Potter TA, Campbell PA. Listeria monocytogenes infects human endothelial cells by two distinct mechanisms. Infect Immun. noviembre de 1995;63(11):4268.
- Faralla C, Rizzuto GA, Lowe DE, Kim B, Cooke C, Shiow LR, et al. InlP, a New Virulence Factor with Strong Placental Tropism. Infect Immun. diciembre de 2016;84(12):3584-96.
- Theriot JA, Rosenblatt J, Portnoy DA, Goldschmidt-Clermont PJ, Mitchison TJ. Involvement of profilin in the actin-based motility of L. monocytogenes in cells and in cell-free extracts. Cell. 11 de febrero de 1994;76(3):505-17.
- Schlech WF. Epidemiology and Clinical Manifestations of Listeria monocytogenes Infection. Microbiol Spectr. 17 de mayo de 2019;7(3):10.1128/microbiolspec.gpp3-0014-2018.
- Janakiraman V. Listeriosis in Pregnancy: Diagnosis, Treatment, and Prevention. Rev Obstet Gynecol. Fall de 2008;1(4):179.
- Ke Y, Ye L, Zhu P, Sun Y, Zhu Z. Listeriosis during pregnancy: a retrospective cohort study. BMC Pregnancy Childbirth. 28 de marzo de 2022;22(1):261.
- Raghupathy R, Szekeres-Bartho J. Progesterone: A Unique Hormone with Immunomodulatory Roles in Pregnancy. Int J Mol Sci. 25 de enero de 2022;23(3):1333.
- Listeriosis [Internet]. [citado 22 de octubre de 2024]. Disponible en: https://www.who.int/news-room/fact-sheets/detail/listeriosis
- Clinical features and prognostic factors of listeriosis: the MONALISA national prospective cohort study - The Lancet Infectious Diseases [Internet]. [citado 22 de octubre de 2024]. Disponible en: https://www.thelancet.com/journals/laninf/article/PIIS1473-3099(16)30521-7/abstract
- Allerberger F, Huhulescu S. Pregnancy related listeriosis: treatment and control. Expert Rev Anti Infect Ther. marzo de 2015;13(3):395-403.
- Frontiers | Different Shades of Listeria monocytogenes: Strain, Serotype, and Lineage-Based Variability in Virulence and Stress Tolerance Profiles [Internet]. [citado 22 de octubre de 2024]. Disponible en: https://www.frontiersin.org/journals/microbiology/articles/10.3389/fmicb.2021.792162/full
- Poyart-Salmeron C, Carlier C, Trieu-Cuot P, Courvalin P, Courtieu AL. Transferable plasmid-mediated antibiotic resistance in Listeria monocytogenes. The Lancet. 16 de junio de 1990;335(8703):1422-6.
- Antimicrobial Resistance in Listeria Species | Microbiology Spectrum [Internet]. [citado 22 de octubre de 2024]. Disponible en: https://journals.asm.org/doi/10.1128/microbiolspec.arba-0031-2017
- Azimi PH, Koranyi K, Lindsey KD. Listeria monocytogenes: Synergistic Effects of Ampicillin and Gentamicin. Am J Clin Pathol. 1 de diciembre de 1979;72(6):974-7.
- Sutter JP, Kocheise L, Kempski J, Christner M, Wichmann D, Pinnschmidt H, et al. Gentamicin combination treatment is associated with lower mortality in patients with invasive listeriosis: a retrospective analysis. Infection. 1 de agosto de 2024;52(4):1601-6.
- Winslow DL, Pankey GA. In vitro activities of trimethoprim and sulfamethoxazole against Listeria monocytogenes. Antimicrob Agents Chemother. julio de 1982;22(1):51.
- Treatment and prevention of Listeria monocytogenes infection - UpToDate [Internet]. [citado 28 de octubre de 2024]. Disponible en: https://www.uptodate.com/contents/treatment-and-prevention-of-listeria-monocytogenes-infection
- Stepanović S, Lazarević G, Jesić M, Kos R. Meropenem therapy failure in Listeria monocytogenes infection. Eur J Clin Microbiol Infect Dis Off Publ Eur Soc Clin Microbiol. junio de 2004;23(6):484-6.
Reviewer 3 Report
Comments and Suggestions for Authors
- In the abstract, it is recommended to add information on the methods used to confirm the diagnosis (e.g., PCR).
- Lines 52–55: This section should be rephrased. The statement "antibiotic therapy can improve fetal outcomes" is too general. Additionally, the phrase "In conclusion" should not be used in the introduction.
- Line 95: "Fluid loss" is not a symptom by itself; it may refer to diarrhea. Alternatively, you could write "symptoms of dehydration" in general.
- It is essential to clarify which specific method was used to identify Listeria. Simply stating "blood cultures were positive" is insufficient. Was a bacteriological method (e.g., culture on esculin agar or blood agar) employed? Or was it PCR? Alternatively, was it a serological method (detection of IgM/IgG)? In Case 2, were there any histopathological signs of listeriosis?
- Line 145: The authors claim that the incidence in this study corresponds to data from the literature. However, the methods and results do not specify the total number of pregnant women included in the study. The text mentions 8 cases per 100,000, but only 6 cases are described. The total number of pregnant women between 2018 and 2023 is not indicated.
- Table 1: All abbreviations should be explained in the notes (e.g., D&C).
- Line 170: It would be helpful to specify the mechanism of immunomodulation under the influence of progesterone. Likely, it involves suppression of the Th1 immune response and other processes.
- Lines 168–169, which state that 80% of infected women experience complications, seem to contradict lines 185–186, which mention asymptomatic forms. This should be clarified.
- Since the manuscript was submitted to the journal "Microorganisms," it is recommended to emphasize microbiological aspects of listeriosis diagnosis.
- The UpToDate protocol (in the “Epidemiology and pathogenesis of Listeria monocytogenes infection” section) indicates that "the highest incidence has been observed in pregnant Hispanic women (7.0 cases per 100,000 population)." As this study was conducted in Spain, it would be relevant to specify the ethnicity of the women included.
- Figure 2: It is important to note that TMP-SMX should be avoided during the first trimester. Furthermore, aside from TMP-SMX, meropenem is also an alternative treatment option (according to UpToDate “Treatment and prevention of Listeria monocytogenes infection”).
- Line 255: in vitro should be written in italics.
- Line 300: There is an incomplete sentence that needs revision.
Author Response
Case Series on Listeria monocytogenes in Pregnancy: Maternal-Fetal Complications and Clinical Management in Six Cases.
Lucía Castaño Frías 1, Carmen Tudela-Littleton Peralta 1, Natalia Segura Oliva 1, María Suárez Arana 1,2,3, Celia Cuenca Marín 1,2,3 and Jesús S. Jiménez López 1,2,3
Response to Reviewer’s comments
|
Thank you very much for taking the time to review this manuscript. Please find the detailed responses below and the corresponding revisions and changes marked in red in the re-submitted files.
|
Point-by-point response to Comments and Suggestions for Authors |
Comment 1: In the abstract, it is recommended to add information on the methods used to confirm the diagnosis (e.g., PCR). |
Response 1: Corrected, thank you. We have added this sentence: Diagnostic confirmation was achieved using methods such as CRP and blood cultures (line 23-24)
|
Comment 2: Lines 52–55: This section should be rephrased. The statement "antibiotic therapy can improve fetal outcomes" is too general. Additionally, the phrase "In conclusion" should not be used in the introduction. |
Response 2: Thank you for your input. We have, accordingly, modified the following paragraph to emphasize this point. Lines 379-385 Regarding treatment, the mothers in our series were treated empirically with ampicillin and gentamicin in accordance with the protocol for the management of chorioamnionitis at our hospital. Neonates were treated with Ampicillin. We are gratified to report that there were no neonatal death. This reinforces the notion that appropriate antibiotic therapy can significantly improve outcomes in cases of neonatal listeriosis (4,19). The therapeutic management outlined in Figure 2 has been implemented with the objective of developing an effective strategy that mitigates the risk of complications for both mother and foetus while optimizing therapeutic efficacy.
Comment 3: Line 95: "Fluid loss" is not a symptom by itself; it may refer to diarrhea. Alternatively, you could write "symptoms of dehydration" in general. Response 3: Thank you for your feedback, we have changed it for the word “amniorrhea” in line 257
Comment 4: It is essential to clarify which specific method was used to identify Listeria. Simply stating "blood cultures were positive" is insufficient. Was a bacteriological method (e.g., culture on esculin agar or blood agar) employed? Or was it PCR? Alternatively, was it a serological method (detection of IgM/IgG)? In Case 2, were there any histopathological signs of listeriosis? Response 4: Thank you. We have included additional information we hope you find satisfactory.
In Line 247 we have written the bacteriological method: ‘different samples cultured on blood agar’. For further detailed information about histopathological signs in stillbirths cases, please refer to the necropsy findings presented in lines 262-263 and 295-297.
Comment 5: Table 1: All abbreviations should be explained in the notes (e.g., D&C). Response 5: Corrected, thank you. We have added the following explanations: Table 1. Summary of Six Clinical Cases of Listeriosis in Pregnant Women (Abbreviations: BC+: Positive Blood Culture; D&C: Dilation and Curettage)
Comment 6: Line 170: It would be helpful to specify the mechanism of immunomodulation under the influence of progesterone. Likely, it involves suppression of the Th1 immune response and other processes. Response 6: Thank you for your feedback. We have added the following information (Lines70-85):
Given that the bacteria exhibit an intracellular life cycle, the host defense against L. monocytogenes relies on cell-mediated immunity, and any condition that weakens this response increases the risk of infection (10,11). The role of immunomodulation during pregnancy is of paramount importance. Progesterone has been demonstrated to inhibit maternal immune responses at the utero-placental interface (6,12). Its mechanisms include the suppression of pro-inflammatory responses, the inhibition of the activation of dendritic cells (DCs), macrophages and natural killer (NK) cells, and the reduction of the production of pro-inflammatory cytokines such as TNF-α and IL-1β. In addition, IL-12, a cytokine that drives Th1 responses associated with inflammation and fetal rejection, is suppressed. Progesterone also inhibits the production of chemokines, including macrophage inflammatory protein-1α, macrophage inflammatory protein-1β and RANTES, by CD8+ T lymphocytes. This reduces the recruitment of immune cells to the placenta and attenuates inflammatory responses. Therefore, while progesterone is essential for maintaining a tolerogenic environment necessary for fetal development, the same immunosuppressive effect may increase susceptibility to infections such as listeriosis (12).
Comment 7: Lines 168–169, which state that 80% of infected women experience complications, seem to contradict lines 185–186, which mention asymptomatic forms. This should be clarified.
Response 7: Thank you for pointing this out. We referred to fetal complications and we have changed it in the manuscript accordingly.
Lines 115-118: It is estimated that over 80% of pregnant women infected with this bacterium experience significant complications for their fetuses or newborns,(4,6,13,14). Moreover, we have clarified the maternal symptoms in lines 121-133: The spectrum of clinical presentations of listeriosis ranges from asymptomatic patients to those presenting with flu-like symptoms such as fever, malaise, myalgias, and mild gastrointestinal symptoms. It may also manifest with non-specific obstetric clinical features, such as uterine contractions, abnormal fetal heart rate, labor, chorioamnionitis, and fetal loss. This nonspecific presentation, common to other conditions, presents a diagnostic challenge to the obstetrician . Maternal symptoms may indicate a higher level of exposure to L. monocytogenes or increased susceptibility to the infection (1,2,5,6). However, about 29% of maternal cases can be asymptomatic, which means that maternal symptoms alone cannot be considered a reliable predictor of adverse fetal effects (15). While listeriosis in pregnant women can be mild, its neonatal consequences can be severe, including spontaneous abortion, preterm birth, and fetal death (1,5,14).
Comment 8: Since the manuscript was submitted to the journal "Microorganisms," it is recommended to emphasize microbiological aspects of listeriosis diagnosis. Response 8: Thank you for your input, we agree and we have included additional information and bibliography that can be found in the introduction section. Lines 40-45 and 154-160.
Comment 9: The UpToDate protocol (in the “Epidemiology and pathogenesis of Listeria monocytogenes infection” section) indicates that "the highest incidence has been observed in pregnant Hispanic women (7.0 cases per 100,000 population)." As this study was conducted in Spain, it would be relevant to specify the ethnicity of the women included. Response 9: Thank you, we agree. Line 246. The following are the clinical cases of six Hispanic pregnant women with confirmed L. monocytogenes infection.
Comment 10: In the discussion section, too much information to be included in the introduction. I ask the authors to change the form of the discussion, i.e., juxtaposing their own results with other clinical cases. Response 10: Thank you for your feedback. Thank you for your feedback. We have removed the bibliographic information from the discussion and included it together with the new information suggested by the reviewers in the introduction. We have also expanded our commentary on the case series in the discussion. We hope you find the revised discussion satisfactory. As so many paragraphs have been changed, the corrections are in all the introduction and discussion.
Comment 11: Figure 2: It is important to note that TMP-SMX should be avoided during the first trimester. Furthermore, aside from TMP-SMX, meropenem is also an alternative treatment option (according to UpToDate “Treatment and prevention of Listeria monocytogenes infection”). Response 11: Thank you for your recommendation. We have followed your suggestion and included additional information on other treatment options, such as in patients allergic to penicillin, in which case TMP-SMX could be an alternative, and in cases of infection during the first trimester and last month of pregnancy, when TMP-SNX should be avoided due to the risk of folic acid metabolism disorders and kernicterus, respectively.
Comment 12: Line 255: in vitro should be written in italics. Response 12: Corrected, thank you.
Comment 13: Line 300: There is an incomplete sentence that needs revision. Response 13: Corrected, thank you. Line 421-425: In this way, early detection and timely treatment become fundamental elements to improve maternal-fetal prognosis and reduce the morbidity and mortality associated with this infection, ultimately leading to better health outcomes for both mothers and their infants.
|
Additional clarifications |
It is our hope that the revisions we have made will be deemed appropriate. The structure of the information contained in the article has been reorganized, with the introduction expanded to include additional information that we felt was relevant to the topic. This includes information regarding the virulence factors of L. monocytogenes, the pathophysiology of the infection and the mechanisms of immunomodulation that occur during pregnancy. The discussion has been focused on comparing our cases with the existing bibliography.
We have included the references below for your consideration:
- 1. Charlier C, Disson O, Lecuit M. Maternal-neonatal listeriosis. Virulence. 31 de diciembre de 2020;11(1):391-7.
- Khsim IEF, Mohanaraj-Anton A, Horte IB, Lamont RF, Khan KS, Jørgensen JS, et al. Listeriosis in pregnancy: An umbrella review of maternal exposure, treatment and neonatal complications. BJOG Int J Obstet Gynaecol. 2022;129(9):1427-33.
- Posfay-Barbe KM, Wald ER. Listeriosis. Semin Fetal Neonatal Med. 1 de agosto de 2009;14(4):228-33.
- Lamont RF, Sobel J, Mazaki-Tovi S, Kusanovic JP, Vaisbuch E, Kim SK, et al. Listeriosis in Human Pregnancy: a systematic review. J Perinat Med. 25 de abril de 2011;39(3):227.
- Moran LJ, Verwiel Y, Bahri Khomami M, Roseboom TJ, Painter RC. Nutrition and listeriosis during pregnancy: a systematic review. J Nutr Sci. 2018;7:e25.
- Craig AM, Dotters-Katz S, Kuller JA, Thompson JL. Listeriosis in Pregnancy: A Review. Obstet Gynecol Surv. junio de 2019;74(6):362-8.
- Drevets DA, Sawyer RT, Potter TA, Campbell PA. Listeria monocytogenes infects human endothelial cells by two distinct mechanisms. Infect Immun. noviembre de 1995;63(11):4268.
- Faralla C, Rizzuto GA, Lowe DE, Kim B, Cooke C, Shiow LR, et al. InlP, a New Virulence Factor with Strong Placental Tropism. Infect Immun. diciembre de 2016;84(12):3584-96.
- Theriot JA, Rosenblatt J, Portnoy DA, Goldschmidt-Clermont PJ, Mitchison TJ. Involvement of profilin in the actin-based motility of L. monocytogenes in cells and in cell-free extracts. Cell. 11 de febrero de 1994;76(3):505-17.
- Schlech WF. Epidemiology and Clinical Manifestations of Listeria monocytogenes Infection. Microbiol Spectr. 17 de mayo de 2019;7(3):10.1128/microbiolspec.gpp3-0014-2018.
- Janakiraman V. Listeriosis in Pregnancy: Diagnosis, Treatment, and Prevention. Rev Obstet Gynecol. Fall de 2008;1(4):179.
- Ke Y, Ye L, Zhu P, Sun Y, Zhu Z. Listeriosis during pregnancy: a retrospective cohort study. BMC Pregnancy Childbirth. 28 de marzo de 2022;22(1):261.
- Raghupathy R, Szekeres-Bartho J. Progesterone: A Unique Hormone with Immunomodulatory Roles in Pregnancy. Int J Mol Sci. 25 de enero de 2022;23(3):1333.
- Listeriosis [Internet]. [citado 22 de octubre de 2024]. Disponible en: https://www.who.int/news-room/fact-sheets/detail/listeriosis
- Clinical features and prognostic factors of listeriosis: the MONALISA national prospective cohort study - The Lancet Infectious Diseases [Internet]. [citado 22 de octubre de 2024]. Disponible en: https://www.thelancet.com/journals/laninf/article/PIIS1473-3099(16)30521-7/abstract
- Allerberger F, Huhulescu S. Pregnancy related listeriosis: treatment and control. Expert Rev Anti Infect Ther. marzo de 2015;13(3):395-403.
- Frontiers | Different Shades of Listeria monocytogenes: Strain, Serotype, and Lineage-Based Variability in Virulence and Stress Tolerance Profiles [Internet]. [citado 22 de octubre de 2024]. Disponible en: https://www.frontiersin.org/journals/microbiology/articles/10.3389/fmicb.2021.792162/full
- Poyart-Salmeron C, Carlier C, Trieu-Cuot P, Courvalin P, Courtieu AL. Transferable plasmid-mediated antibiotic resistance in Listeria monocytogenes. The Lancet. 16 de junio de 1990;335(8703):1422-6.
- Antimicrobial Resistance in Listeria Species | Microbiology Spectrum [Internet]. [citado 22 de octubre de 2024]. Disponible en: https://journals.asm.org/doi/10.1128/microbiolspec.arba-0031-2017
- Azimi PH, Koranyi K, Lindsey KD. Listeria monocytogenes: Synergistic Effects of Ampicillin and Gentamicin. Am J Clin Pathol. 1 de diciembre de 1979;72(6):974-7.
- Sutter JP, Kocheise L, Kempski J, Christner M, Wichmann D, Pinnschmidt H, et al. Gentamicin combination treatment is associated with lower mortality in patients with invasive listeriosis: a retrospective analysis. Infection. 1 de agosto de 2024;52(4):1601-6.
- Winslow DL, Pankey GA. In vitro activities of trimethoprim and sulfamethoxazole against Listeria monocytogenes. Antimicrob Agents Chemother. julio de 1982;22(1):51.
- Treatment and prevention of Listeria monocytogenes infection - UpToDate [Internet]. [citado 28 de octubre de 2024]. Disponible en: https://www.uptodate.com/contents/treatment-and-prevention-of-listeria-monocytogenes-infection
- Stepanović S, Lazarević G, Jesić M, Kos R. Meropenem therapy failure in Listeria monocytogenes infection. Eur J Clin Microbiol Infect Dis Off Publ Eur Soc Clin Microbiol. junio de 2004;23(6):484-6.
Reviewer 4 Report
Comments and Suggestions for Authors
This manuscript, entitled “Impact of Listeria monocytogenes in Pregnancy: Maternal-Fetal Complications and Diagnostic-Therapeutic Management” had described the 6 confirmed cases of gestational listeriosis at Hospital Materno Infantil de Málaga, focusing on clinical presentations, management, and outcomes to enhance understanding and improve diagnosis and treatment.
Overall, the manuscript is well organized, however, as indicated by the authors, the lack of a control group and the small sample size restrict the generalizability of the findings. These are very big limitations which cannot be ignored. Thus, I suggest the authors include control groups (i.e. healthy pregnant women in the same hospital, non-pregnant individuals with gestational listeriosis) and larger the sample size (considering such case is not common, this one can be compromised) to enhance the convincement of the results.
Concerning literature review, the search criteria included keywords such as “Listeria monocytogenes,” “infection in pregnant women,” and “maternal-fetal complications.” I suggest expanding the searched keywords to get more relevant studies. For example, Listeria monocytogenes, pregnant; Listeria monocytogenes, maternal; etc. Making the keywords simple could help getting more results.
Some writing and formatting issues should be revised. For example, the “Listeria monocytogenes” in title and keywords should be italic. “2. Materiales y Métodos” should be written in English.
Comments on the Quality of English LanguageSome writing and formatting issues should be revised. For example, the “Listeria monocytogenes” in title and keywords should be italic. “2. Materiales y Métodos” should be written in English.
Author Response
Case Series on Listeria monocytogenes in Pregnancy: Maternal-Fetal Complications and Clinical Management in Six Cases.
Lucía Castaño Frías 1, Carmen Tudela-Littleton Peralta 1, Natalia Segura Oliva 1, María Suárez Arana 1,2,3, Celia Cuenca Marín 1,2,3 and Jesús S. Jiménez López 1,2,3
Response to Reviewer’s comments
|
Thank you very much for taking the time to review this manuscript. Please find the detailed responses below and the corresponding revisions and changes marked in red in the re-submitted files.
|
Point-by-point response to Comments and Suggestions for Authors |
Comment 1: Overall, the manuscript is well organized, however, as indicated by the authors, the lack of a control group and the small sample size restrict the generalizability of the findings. These are very big limitations which cannot be ignored. Thus, I suggest the authors include control groups (i.e. healthy pregnant women in the same hospital, non-pregnant individuals with gestational listeriosis) and larger the sample size (considering such case is not common, this one can be compromised) to enhance the convincement of the results. Response 1: Thank you for your feedback and recommendations to improve our study. We understand and acknowledge the limitations associated with the lack of a control group and the small sample size, as noted in the manuscript. However, as this study is designed as a case series, the introduction of a control group would significantly alter the study design and objectives. Given the rarity of gestational listeriosis cases, obtaining a suitable control group and increasing the sample size in our setting would be very challenging. Our primary aim was to provide a detailed description and analysis of the observed cases and, despite these limitations, to contribute to the existing knowledge base. We hope this clarification is helpful and appreciate your understanding of the scope and limitations of the study.
Comment 2: Concerning literature review, the search criteria included keywords such as “Listeria monocytogenes,” “infection in pregnant women,” and “maternal-fetal complications.” I suggest expanding the searched keywords to get more relevant studies. For example, Listeria monocytogenes, pregnant; Listeria monocytogenes, maternal; etc. Making the keywords simple could help getting more results. Response 2: Thank you. We have expanded the bibliography by including the keywords “diagnosis” and “treatment” in our literature search. This allowed us to include more relevant studies and improve the comprehensiveness of our review.
Comment 3: Comments on the Quality of English Language. Some writing and formatting issues should be revised. For example, the “Listeria monocytogenes” in title and keywords should be italic. “2. Materiales y Métodos” should be written in English. Response 3: Corrected, thank you. We have written Listeria monocytogenes in italics, and we have also changed that section heading to “Materials and Methods” in English (line 223)
|
|
Additional clarifications |
It is our hope that the revisions we have made will be deemed appropriate. The structure of the information contained in the article has been reorganized, with the introduction expanded to include additional information that we felt was relevant to the topic. This includes information regarding the virulence factors of L. monocytogenes, the pathophysiology of the infection and the mechanisms of immunomodulation that occur during pregnancy. The discussion has been focused on comparing our cases with the existing bibliography.
We have included the references below for your consideration:
- 1. Charlier C, Disson O, Lecuit M. Maternal-neonatal listeriosis. 31 de diciembre de 2020;11(1):391-7.
- Khsim IEF, Mohanaraj-Anton A, Horte IB, Lamont RF, Khan KS, Jørgensen JS, et al. Listeriosis in pregnancy: An umbrella review of maternal exposure, treatment and neonatal complications. BJOG Int J Obstet Gynaecol. 2022;129(9):1427-33.
- Posfay-Barbe KM, Wald ER. Listeriosis. Semin Fetal Neonatal Med. 1 de agosto de 2009;14(4):228-33.
- Lamont RF, Sobel J, Mazaki-Tovi S, Kusanovic JP, Vaisbuch E, Kim SK, et al. Listeriosis in Human Pregnancy: a systematic review. J Perinat Med. 25 de abril de 2011;39(3):227.
- Moran LJ, Verwiel Y, Bahri Khomami M, Roseboom TJ, Painter RC. Nutrition and listeriosis during pregnancy: a systematic review. J Nutr Sci. 2018;7:e25.
- Craig AM, Dotters-Katz S, Kuller JA, Thompson JL. Listeriosis in Pregnancy: A Review. Obstet Gynecol Surv. junio de 2019;74(6):362-8.
- Drevets DA, Sawyer RT, Potter TA, Campbell PA. Listeria monocytogenes infects human endothelial cells by two distinct mechanisms. Infect Immun. noviembre de 1995;63(11):4268.
- Faralla C, Rizzuto GA, Lowe DE, Kim B, Cooke C, Shiow LR, et al. InlP, a New Virulence Factor with Strong Placental Tropism. Infect Immun. diciembre de 2016;84(12):3584-96.
- Theriot JA, Rosenblatt J, Portnoy DA, Goldschmidt-Clermont PJ, Mitchison TJ. Involvement of profilin in the actin-based motility of L. monocytogenes in cells and in cell-free extracts. Cell. 11 de febrero de 1994;76(3):505-17.
- Schlech WF. Epidemiology and Clinical Manifestations of Listeria monocytogenes Infection. Microbiol Spectr. 17 de mayo de 2019;7(3):10.1128/microbiolspec.gpp3-0014-2018.
- Janakiraman V. Listeriosis in Pregnancy: Diagnosis, Treatment, and Prevention. Rev Obstet Gynecol. Fall de 2008;1(4):179.
- Ke Y, Ye L, Zhu P, Sun Y, Zhu Z. Listeriosis during pregnancy: a retrospective cohort study. BMC Pregnancy Childbirth. 28 de marzo de 2022;22(1):261.
- Raghupathy R, Szekeres-Bartho J. Progesterone: A Unique Hormone with Immunomodulatory Roles in Pregnancy. Int J Mol Sci. 25 de enero de 2022;23(3):1333.
- Listeriosis [Internet]. [citado 22 de octubre de 2024]. Disponible en: https://www.who.int/news-room/fact-sheets/detail/listeriosis
- Clinical features and prognostic factors of listeriosis: the MONALISA national prospective cohort study - The Lancet Infectious Diseases [Internet]. [citado 22 de octubre de 2024]. Disponible en: https://www.thelancet.com/journals/laninf/article/PIIS1473-3099(16)30521-7/abstract
- Allerberger F, Huhulescu S. Pregnancy related listeriosis: treatment and control. Expert Rev Anti Infect Ther. marzo de 2015;13(3):395-403.
- Frontiers | Different Shades of Listeria monocytogenes: Strain, Serotype, and Lineage-Based Variability in Virulence and Stress Tolerance Profiles [Internet]. [citado 22 de octubre de 2024]. Disponible en: https://www.frontiersin.org/journals/microbiology/articles/10.3389/fmicb.2021.792162/full
- Poyart-Salmeron C, Carlier C, Trieu-Cuot P, Courvalin P, Courtieu AL. Transferable plasmid-mediated antibiotic resistance in Listeria monocytogenes. The Lancet. 16 de junio de 1990;335(8703):1422-6.
- Antimicrobial Resistance in Listeria Species | Microbiology Spectrum [Internet]. [citado 22 de octubre de 2024]. Disponible en: https://journals.asm.org/doi/10.1128/microbiolspec.arba-0031-2017
- Azimi PH, Koranyi K, Lindsey KD. Listeria monocytogenes: Synergistic Effects of Ampicillin and Gentamicin. Am J Clin Pathol. 1 de diciembre de 1979;72(6):974-7.
- Sutter JP, Kocheise L, Kempski J, Christner M, Wichmann D, Pinnschmidt H, et al. Gentamicin combination treatment is associated with lower mortality in patients with invasive listeriosis: a retrospective analysis. Infection. 1 de agosto de 2024;52(4):1601-6.
- Winslow DL, Pankey GA. In vitro activities of trimethoprim and sulfamethoxazole against Listeria monocytogenes. Antimicrob Agents Chemother. julio de 1982;22(1):51.
- Treatment and prevention of Listeria monocytogenes infection - UpToDate [Internet]. [citado 28 de octubre de 2024]. Disponible en: https://www.uptodate.com/contents/treatment-and-prevention-of-listeria-monocytogenes-infection
- Stepanović S, Lazarević G, Jesić M, Kos R. Meropenem therapy failure in Listeria monocytogenes infection. Eur J Clin Microbiol Infect Dis Off Publ Eur Soc Clin Microbiol. junio de 2004;23(6):484-6.
Round 2
Reviewer 1 Report
Comments and Suggestions for Authors
I thank the authors for making the corrections. I accept the article in its present form.
Author Response
Case Series on Listeria monocytogenes in Pregnancy: Maternal-Fetal Complications and Clinical Management in Six Cases.
Lucía Castaño Frías 1, Carmen Tudela-Littleton Peralta 1, Natalia Segura Oliva 1, María Suárez Arana 1,2,3, Celia Cuenca Marín 1,2,3 and Jesús S. Jiménez López 1,2,3
Response to Reviewer’s comments
Dear editor,
I would like to express our sincere gratitude for your thoughtful and constructive feedback on our manuscript. We greatly appreciate the time and effort that went into reviewing the submission, and we found the comments to be extremely valuable in improving the quality of our work. We have carefully considered all your suggestions and have revised the manuscript accordingly.
Reviewer 2 Report
Comments and Suggestions for Authors
The authors present a (small) series of clincal cases o listerioses in newborn. The information given is not really new. Hence this paper does not add any new information on the topic.
The microbilogical diagnosis is made by classical methods : PCR would possibly have revealed even more cases.
By the way, actually (in 2024) 28 species of Listeria are described.
Author Response
Case Series on Listeria monocytogenes in Pregnancy: Maternal-Fetal Complications and Clinical Management in Six Cases.
Lucía Castaño Frías 1, Carmen Tudela-Littleton Peralta 1, Natalia Segura Oliva 1, María Suárez Arana 1,2,3, Celia Cuenca Marín 1,2,3 and Jesús S. Jiménez López 1,2,3
Response to Reviewer’s comments
Dear editor,
I would like to express our sincere gratitude for your thoughtful and constructive feedback on our manuscript. We greatly appreciate the time and effort that went into reviewing the submission, and we found the comments to be extremely valuable in improving the quality of our work. We have carefully considered all your suggestions and have revised the manuscript accordingly.
We recognise that our series of cases may be relatively short due to the low prevalence of the infection in our region. However, we believe it is valuable to review, update and present diagnostic algorithms that could be employed in a straightforward manner within the context of daily clinical practice
Reviewer 3 Report
Comments and Suggestions for Authors
My comments were taken into account. Thank you.
Author Response

(The authors gave the same response as above.)

Reviewer 4 Report
Comments and Suggestions for Authors
I appreciate the response and revision from the authors. The manuscript has been improved. However, in the revision, the introduction is too lengthy, please shorten it to be concise.
Author Response
Case Series on Listeria monocytogenes in Pregnancy: Maternal-Fetal Complications and Clinical Management in Six Cases.
Lucía Castaño Frías 1, Carmen Tudela-Littleton Peralta 1, Natalia Segura Oliva 1, María Suárez Arana 1,2,3, Celia Cuenca Marín 1,2,3 and Jesús S. Jiménez López 1,2,3
Response to Reviewer’s comments
Dear editor,
I would like to express our sincere gratitude for your thoughtful and constructive feedback on our manuscript. We greatly appreciate the time and effort that went into reviewing the submission, and we found the comments to be extremely valuable in improving the quality of our work. We have carefully considered all your suggestions and have revised the manuscript accordingly. Nevertheless, the introduction was extended to accommodate the additional suggestions put forth by the other reviewers.